# Autoimmune Encephalitis and Paraneoplastic Neurological Syndromes with Progressive Supranuclear Palsy-like Manifestations

**DOI:** 10.3390/brainsci14101012

**Published:** 2024-10-09

**Authors:** Naoki Yamahara, Akira Takekoshi, Akio Kimura, Takayoshi Shimohata

**Affiliations:** Department of Neurology, Gifu University Graduate School of Medicine, 1-1 Yanagido, Gifu 501-1194, Japan; naoki_yamahara@icloud.com (N.Y.); takekoshi.akira.f3@f.gifu-u.ac.jp (A.T.); kimura.akio.k8@f.gifu-u.ac.jp (A.K.)

**Keywords:** progressive supranuclear palsy, autoimmune encephalitis, paraneoplastic neurological syndrome

## Abstract

Background: Advances in diagnostic procedures have led to an increasing rate of diagnosis of autoimmune encephalitis or paraneoplastic neurological syndrome (AE/PNS) among patients with progressive supranuclear palsy (PSP)-like manifestations. Methods: In this narrative review, we first discuss the clinical characteristics of AE/PNS in comparison to those of PSP, followed by a discussion of diagnosis and treatment. Results: The antibodies involved in these conditions include anti-IgLON5, -Ma2, and -Ri antibodies, each of which has a characteristic clinical presentation. The steps in the diagnosis of AE/PNS in patients with PSP-like manifestations include (i) suspicion of AE/PNS based on clinical presentations atypical of PSP and (ii) antibody detection measures. Methods used to identify antibodies include a combination of tissue-based assays and confirmatory tests. The primary confirmatory tests include cell-based assays and immunoblotting. Treatments can be divided into immunotherapy and tumor therapies, the former of which includes acute and maintenance therapies. Conclusions: One of the major challenges of diagnosis is that existing reports on PSP-like patients with AE/PNS include only case reports, with the majority discussing antibodies other than anti-IgLON5 antibody. As such, more patients need to be evaluated to establish the relationship between antibodies and PSP-like manifestations.

## 1. Introduction

Progressive supranuclear palsy (PSP), first described by Steele, Richardson, and Olszewski in 1964, is believed to be caused by tauopathy, where gain of toxic function is caused by tau aggregates formed through aberrant post-translational modifications, such as excessive phosphorylation, and loss of function occurs due to a reduction in normal tau function [1]. Tauopathies are classified into three-repeat (3R) tauopathies, which have three repeat sequences in the microtubular-binding domain of tau protein, four-repeat (4R) tauopathies, which have four, and 3R/4R tauopathies, which have both. PSP is a representative disease of 4R tauopathies [2]. PSP typically presents with vertical supranuclear gaze palsy, repeated falls within 3 years, a more rapid progression than Parkinson’s disease, and a resistance to levodopa therapy; however, attention is necessary as multiple clinical variants have been reported [3,4]. Clinical diagnosis is currently made using the Movement Disorder Society clinical diagnostic criteria for PSP (MDS-PSP criteria) (Table 1) [5]. As diseases that present with PSP syndrome are diverse, including hereditary disorders, prion diseases, cerebrovascular diseases, infections, and, notably, autoimmune disorders, thorough differential diagnosis is crucial [6,7]. The gold standard for diagnosis of PSP is a pathological diagnosis, in which two criteria are currently used [8,9]. There is no established effective treatment, and management primarily consists of levodopa trials and rehabilitation.

Recently, the importance of autoimmune encephalitis (AE) and paraneoplastic neurological syndrome (PNS) as mimics of Parkinsonian syndromes, including PSP, has been increasingly recognized [10,11]. The pathogenesis of AE/PNS is thought to be divided into two types: humoral- and cellular-immunity-driven [12]. Pathogenic autoantibodies are present in disorders predominantly caused by humoral immunity. Autoantibodies can also exist in many cellular-immunity-mediated cases, although most are considered secondary occurrences without pathological significance. Over the past decade, advances in antibody identification techniques, such as cell-based assays (CBA), immunoprecipitation methods, and protein microarrays, have led to the discovery of many novel autoantibodies that can cause AE/PNS, some of which have been associated with symptoms and signs that imitate PSP. Although pathophysiological similarities of PSP and AE/PNS have not been elucidated, neuroinflammation is increasingly recognized as an important aspect of PSP. Site and intensity of neuroinflammation were correlated with tau pathology [13] and clinical severity [14] on positron emission tomography; and cerebrospinal fluid (CSF) cytokines suggesting microglial activation including tumor necrosis factor alpha, interleukin (IL)-1β, and IL-6 were elevated in PSP patients compared to controls [15].

As AE/PNS can be treated with immunotherapies and tumor therapies, clinicians should differentiate between these conditions. In this review, we discuss mainly the following topics: (i) clinical features of AE/PNS mimicking PSP in comparison to PSP and (ii) the diagnosis and management of PSP-like AE/PNS.

## 2. Methods

We performed a narrative review of articles retrieved from the PubMed database (search date: 18 June 2024) using the following search string: ((progressive supranuclear palsy [MeSH Terms] OR “progressive supranuclear pals*” [Title/Abstract] OR PSPRS [Title/Abstract] OR “Richardson* syndrome” [Title/Abstract] OR “progressive gait freezing” [Title/Abstract] OR “pure akinesia with gait freezing” [Title/Abstract]) AND (antibod*[Title/Abstract] OR autoantibod*[Title/Abstract] OR immunolog*[Title/Abstract] OR autoimmune*[Title/Abstract] OR immune*[Title/Abstract] OR neoplas*[Title/Abstract] OR paraneoplas*[Title/Abstract])) AND ((english[Language]) OR (japanese[Language])). The inclusion period was unlimited. A total of 334 articles matched the search criteria, from which 37 case reports, case series, clinical studies, reviews, and basic research articles regarding AE/PNS causing PSP-like manifestations (at least one of the following: vertical supranuclear gaze palsy, frequent falls, postural instability, or gait freezing, which are not clearly explained by other etiologies; or the decision of these authors if details are unclear) were selected. Furthermore, an additional 32 articles were identified by manual searching and referring to bibliographies.

## 3. Autoantibodies Related to AE/PNS Mimicking PSP

### 3.1. Overview of Autoantibodies Mimicking PSP

Since the early 2000s, a series of AE/PNS patients with PSP-like phenotypes have been reported (Table 2). We found 24 actual PSP-like patients in 15 studies (one case series and fourteen case reports).

Throughout all the studies, there are three key points. The first is antibody type. The most discussed condition resembling PSP is anti-IgLON5 disease, a disorder associated with the anti-IgLON5 antibody, with ten patients with PSP-like manifestations examined in one case series [16] and an additional two patients presented in two case reports. The second most notable condition is associated with anti-Ma2 and anti-Ri antibodies. Although actual patients resembling those with PSP have been discussed only in case reports (two were related to anti-Ma2 antibodies, one was regarding anti-Ri antibodies), clinical studies have shown that symptoms and findings can be somewhat similar to those of PSP. In addition to these antibodies, one case report presented a patient mimicking PSP with each of the following antibodies: anti-CV2/collapsin response mediator protein 5 (CV2/CRMP5), anti-Hu, anti-dipeptidyl-peptidase-like protein 6 (DPPX), anti-kelch-like protein 11 (KLHL11), anti-leucine-rich glioma-inactivated 1 protein (LGI1), anti-Sez1l6 antibodies. Uncharacterized antibodies and anti-NH2 terminus of alpha-enolase (NAE) antibodies have also been detected in PSP-mimicking AE/PNS [17,18,19,20].

The second is overall clinical features. Among the 24 patients, their ages ranged from 36 to 79, with eight women and sixteen men. The progression in patients with sufficient information was subacute (10/22) and chronic (12/22). Coexisting tumors included small-cell lung cancer (SCLC) (2/14), breast cancer (2/14), testicular cancer (1/14), and nothing (9/14). The main PSP-like features included vertical gaze palsy (17/23), postural instability (19/23), and gait freezing (3/23). Frequent manifestations that differed from those of PSP included CSF abnormalities, such as elevated protein or cell counts and positive CSF-specific oligoclonal bands (OCBs) (8/12), no atrophy of midbrain tegmentum on MRI (6/24), weight loss (3/24: associated with anti-DPPX, anti-CV2/CRMP5, and anti-KLHL11 antibody in one case each), and sleep disturbance and limb ataxia (3/24: all associated with the anti-IgLON5 antibody). Treatments including corticosteroids (8/12), intravenous immunoglobulin (IVIG) (7/12), rituximab (RTX) (3/12), cyclophosphamide (1/12), and tumor therapy (chemotherapy (4) and radiotherapy (1)) were effective (7/12), temporarily effective (1/12), and ineffective (4/12).

Third, the strength of these studies lies in their ability to examine detailed clinical information; however, most of them are case reports, raising concerns about reproducibility.

In the subsequent sections, we discuss two key points for each antibody. The first subsection addresses the general aspects of diseases associated with each antibody, without focusing on the patients presenting with PSP-like symptoms. The second subsection then discusses the role of the antibodies in PSP.

### 3.2. Anti-IgLON5 Antibody

#### 3.2.1. General Features of Anti-IgLON5 Antibody

IgLON5 is a cell surface antigen primarily expressed in the brain tissue, which is related to the development and formation of neural networks [21]. Anti-IgLON5 disease manifests as low-risk (10–15%) tumor association. Related tumors include breast, thyroid, and kidney cancers [21]. Approximately 75% of the cases have a chronic course, while the remaining 25% have an acute or subacute course. Typical patients have a combination of the following symptoms: (1) bulbar symptoms, (2) sleep disturbances, (3) movement disorders (primarily gait disturbance, generalized chorea, and craniofacial dyskinesias), (4) neuromuscular symptoms (primarily fasciculations, weakness, and stiff-person syndrome), (5) cognitive decline, (6) gaze palsy and blepharoptosis, and (7) autonomic dysfunctions [22]. In suspicious cases, the anti-IgLON5 antibody was measured by CBA. Both serum and CSF analyses are sensitive [23]. Immunotherapies are occasionally successful, with response rates ranging from 10 to 60% [23,24,25,26].

**Table 2 brainsci-14-01012-t002:** Summary of previously published cases of PSP-like patients due to AE/PNS.

Related Ab.	Ag. Type	Sample with Positive Ab.	N	Age	Sex	Progression Pattern	Tumor	PSP-like Manifestations	Atypical Features	Abn. CSF Findings	Immunotherapies and/or Cancer Treatments	Outcome	Reference
DPPX	S	Serum *	1	36	F	NA	None	Initially misdiagnosed with PSP (details unknown)	Diarrhea, weight loss (45 kg), and dysgeusia, myoclonus	NA	Corticosteroids	Not effective	Tobin et al., 2014 [27]
IgLON5	S	CSF	1	66	M	Chronic	None	Gait akinesia (I) and VGP (3 y)	Hypersomnia (I), facial myokymia and myorhythmia (3 y), no atrophy of midbrain tegmentum on MRI (3 y)	NA	Not done	NA	González-Ávila et al., 2020 [28]
IgLON5	S	Serum *	10	62 (44–71) ^†^	M = 7F = 3	Chronic	NA	VGP (8), frequent falls (10), PI (10), and gait akinesia (10)	Sleep dysfunction (3), limb ataxia (3), OH (1), vocal cord palsy (2), respiratory failure (2), limb stiffness with spasms (1), and chorea (1)	NA	NA	NA	Gaig et al., 2021 [16]
IgLON5	S	CSF	1	67	M	Chronic	None	Gait akinesia (I), PI (8 y), depression, downward slow saccades (8 y), and pathology showing 4-repat tauopathy and tufted astrocytes (8.3 y)	Diplopia (I), severe dysphagia with vocal cord paresis (8 y), and nocturnal stridor with severe hypoxemic episodes (8 y)	Normal	Not done	NA	Berger-Sieczkowski et al., 2023 [29]
LGI1	S	Serum	1	60	M	Subacute	None	Frontal dementia (I), PI (6 m) (wheel-chair bound in 4 y), supranuclear VGP (12 m), rigidity (12 m), and retrocollis	Parinaud’s syndrome (12 m), and no atrophy of midbrain tegmentum on MRI (12 m)	Cells and proteins	IVIG, corticosteroids, RTX	Effective	Hierro et al., 2020 [30]
NAE	S	Serum	1	63	F	Subacute	None	Gait akinesia (I) and PI (10 m),	Extremity edema (I), arthralgia (I), no rigidity in the neck, and no atrophy of midbrain tegmentum on MRI (10 m)	Cells and proteins	Corticosteroids	Effective	Inoue et al., 2012 [19]
Sez6l2	S	Serum	1	55	F	Subacute	None	PI (I) (bed bound in 9 m), and gait akinesia (I)	Prominent cerebellar ataxia (9 m) and no atrophy of midbrain tegmentum on MRI (4 m)	Normal	IVIG, RTX	Effective	Borsche et al., 2019 [31]
CV2/ CRMP5	I	Serum, CSF	1	65	M	Subacute	SCLC	Frontal dementia (I) (bed-bound in 4 m), PI (6 m), supranuclear VGP (6 m), and limb rigidity (6 m)	Weight loss (I) (10 kg over 6 months) and T2H of the basal ganglia in MRI (6 m)	Cells	IVIG, chemotherapy	Effective	Dash et al., 2016 [32]
Hu	I	Serum	1	57	M	Subacute	SCLC	Supranuclear VGP (I) and cognitive dysfunction (3 w)	T2 hyperintensity in the extreme capsule on MRI (3 w)	Proteins	IVIG, chemotherapy, radiotherapy	Not effective	Ohyagi et al., 2017 [33]
KLHL11	I	NA	1	79	M	NA	None	VGP and PI	Weight loss (9 kg over 2 months), hypersomnia, and T2H of the mesial temporal region in MRI	Cells, proteins, and OCB	IVIG, corticosteroids	Not effective	Vogrig et al., 2021 [34]
Ma1, Ma2	I	Serum	1	55	M	Subacute	TC	VGP (1 m), FOG (3 m), and urinary urgency (3 m)	Hyperphagia (I), diplopia (2 m), and narcolepsy with cataplexy (3 m)	Proteins	Corticosteroids, CYC	Effective	Adams et al., 2011 [35]
Ma2	I	Serum	1	49	F	Subacute	Breast cancer	Gait akinesia (I) (bed-bound in 1y), PI (I), VGP (1y), and axial rigidity	None	OCB	Corticosteroids, RTX	Effective	Sankhla et al., 2024 [36]
Ri	I	Serum, CSF	1	45	F	Subacute	Breast cancer	FOG (I) (bed-bound in 5 m), PI (6 m), and vertical/horizontal gaze palsy (6 m)	Leg spasticity (6 m) and no atrophy of midbrain tegmentum on MRI (6 m)	Normal	IVIG, chemotherapy	Temporarily effective	Takkar et al., 2020 [37]
UA	NA	NA	1	66	M	Subacute	None	Cognitive dysfunction (I), PI (I), and VGP (6 m)	No atrophy of midbrain tegmentum on MRI (6 m)	Proteins	Corticosteroids	Effective	Kannoth et al., 2016 [17]
UA	NA	CSF	1	72	M	Subacute	None	FOG (I), PI (I), cognitive dysfunction, and upward gaze palsy (1.5 y)	None	Proteins and OCB	IVIG, corticosteroids	Not effective (dead 2 years after onset)	Dale et al., 2018 [18]

The descriptions (I), (3 y), etc., indicate the timing of onset. For example, (I) refers to the initial symptoms, while (3 y) indicates occurrence three years after onset. * Done with CSF but results unavailable. ^†^ Represented by the median. Abbreviations: antibody, Ab.; abnormal, Abn.; antigen, Ag.; cerebrospinal fluid, CSF; CV2/collapsin response mediator protein 5, CV2/CRMP5; cyclophosphamide, CYC; dipeptidyl-peptidase-like protein 6, DPPX; freezing of gait, FOG; intracellular, I; intravenous immunoglobulin, IVIG; Kelch-like protein 11, KLHL11; leucine-rich glioma-inactivated 1, LGI1; magnetic resonance imaging, MRI; not applicable, NA; NH2 terminal of alpha-enolase, NAE; oligoclonal band, OCB; postural instability, PI; progressive supranuclear palsy, PSP; rituximab, RTX; cell surface, S; small-cell lung cancer, SCLC; T2 hyperintensity, T2H; tonsillar carcinoma, TC; uncharacterized antibody, UA; vertical gaze palsy, VGP.

#### 3.2.2. The Role of Anti-IgLON5 Antibody in PSP-like Manifestations

Three key points regarding anti-IgLON5 antibodies must be addressed:

First, the associations between PSP-like syndromes and anti-IgLON5 disease have been discussed more frequently than those with other antibodies [16,28]. There is only one case series study conducting a case-by-case analysis of patients with PSP-like manifestations in this disease [16]. In this study, 27 patients with anti-IgLON5 disease had movement disorders as the main symptom. Ten of the twenty-seven patients presented with signs and symptoms of PSP (gait disturbance (10), postural instability and frequent falls (10), and vertical gaze palsy (8)). Nine patients resembled PSP-Richardson syndrome, whereas one resembled PSP-corticobasal syndrome, with symptoms including vertical gaze palsy, bilateral limb and orobuccal apraxia, asymmetric alien limb phenomenon, a rigid-akinetic syndrome, and mild finger myoclonus. All patients had a chronic disease course, with a median time to diagnosis of 96 months. Moreover, it is important to discuss the symptoms and signs that occur in anti-IgLON5 disease but are not present in PSP. These included minimal downward gaze palsy and prominent sleep disturbances, including abnormal behavior, limb ataxia, orthostatic hypotension, and stridor due to vocal cord paralysis.

Second, the probability of patients clinically diagnosed with PSP testing positive for anti-IgLON5 antibodies is very low. In one study of 33 patients clinically diagnosed with PSP, Mangesius et al. reported that not a single patient tested positive for anti-IgLON5 antibodies [38]. Sabater et al. showed that only one of thirty-two patients with PSP tested positive for anti-IgLON5 antibodies [26], suggesting that it is impractical to measure IgLON5 antibodies in all patients clinically diagnosed with PSP.

Third, anti-IgLON5 antibodies may cause tau pathology [29]. Growing evidence has indicated that these antibodies frequently cause 3-repeat and 4-repeat tauopathy in the later stages, which is considered a secondary phenomenon [22]. Moreover, 4-repeat tauopathy was also found to be present alone in a patient with tufted astrocytes, a pathological hallmark of PSP; however, these features were reported only in a single patient [9]. In this patient, the only finding considered atypical of PSP was the presence of neurofibrillary tangles and neuropil threads, abnormal tau accumulated mainly in distal dendrites, in the synaptic glomeruli of the cerebellar cortex.

Overall, while there is a substantial proportion of patients with PSP syndrome among those with anti-IgLON5 disease, there are very few patients with anti-IgLON5 disease among those presenting with PSP syndrome. Furthermore, anti-IgLON5 disease may cause tauopathy and might present with pathologically defined PSP in some cases.

### 3.3. Anti-Ma2 Antibodies

#### 3.3.1. General Features of Anti-Ma2 Antibody

Ma2 is an intracellular antigen that is normally expressed in the brain tissue, and there is an important isoform called Ma1. Notably, in most cases where autoantibodies are pathogenic, anti-Ma2 antibodies alone or both anti-Ma2 and -Ma1 antibodies are positive. Anti-Ma1 antibodies are rarely detected alone.

Diseases associated with the anti-Ma2 antibody are at a high risk (>75%) of tumor co-existence, which primarily includes testicular tumors and non-small-cell lung cancer (NSCLC) [39]. Typical clinical presentations include subacute limbic encephalitis (e.g., amnesia, seizures, and disturbance of consciousness), brainstem encephalitis (e.g., oculomotor dysfunction, dysarthria, dysphagia, and parkinsonism), and narcolepsy [40]. In suspected patients, anti-Ma2 antibodies can be measured by immunoblot [41,42,43]. The response to immunotherapy and tumor treatment is moderate, and may be superior to that of other diseases associated with antibodies against intracellular antigens. Dalmau et al. showed that 14 of 17 patients treated with tumor treatment (nine of whom were also treated with immunotherapy) improved, while four of ten patients receiving only immunotherapy responded [40].

#### 3.3.2. The Role of Anti-Ma2 Antibody in PSP-like Manifestations

There have been two case reports describing actual PSP-like patients in detail, while one clinical study has shown that at least some of the manifestations caused by anti-Ma2 antibodies can masquerade as PSP. In the two existing case reports, the first patient was a 55-year-old male with no tumors and subacute progressive disease. The initial symptom was hyperphagia. Over the following three months, typical PSP features such as vertical gaze palsy, gait freezing, and urinary urgency developed, along with atypical symptoms including diplopia and narcolepsy with cataplexy [35]. Sankhla et al. also reported a case of a 49-year-old woman with breast cancer with a subacute course. The patient initially presented with gait akinesia and postural instability, resulting in her being bed-bound within a year. Vertical gaze palsy appeared one year after onset. An unusual finding for PSP was the detection of CSF-specific OCBs [36]. Immunotherapy was effective in both of these patients. In another study containing 38 patients with symptoms associated with the anti-Ma2 antibody, twelve had vertical gaze palsy, twelve had excessive daytime sleepiness, five had unsteady gait or mild ataxia, and three had parkinsonism [40].

### 3.4. Anti-Ri (ANNA-2) Antibody

#### 3.4.1. General Features of Anti-Ri Antibody

The anti-Ri antibody binds to two intracellular antigens, the neuro-oncologic ventral antigens (NOVA) 1 and 2, which are expressed in tissues throughout the body, and are involved in the regulation of the alternative splicing of pre-mRNAs. Diseases related to anti-Ri antibodies are commonly associated with tumors (>70%), of which breast cancer is the most frequently observed, followed by lung cancer (including NSCLC and SCLC) [44]. They show a female predominance. Historically, anti-Ri antibody was known as the causative antibody of opsoclonus–myoclonus syndrome (OMS); however, it subsequently became clear that this antibody is associated with a variety of clinical manifestations. The typical presentation is subacute and stepwise or chronic and progressive cerebellar ataxia (which is the most common), oculomotor dysfunction, opsoclonus with/without myoclonus, spasticity, dystonia, and parkinsonism [45]. Immunotherapy and tumor therapy have limited benefits [45,46].

#### 3.4.2. The Role of Anti-Ri Antibody in PSP-like Manifestations

Although only one case report described an actual patient with PSP, a larger case series showed that signs and symptoms of the disease associated with anti-Ri antibodies can also occur in PSP, at least in part [45]. In one review of 36 patients, six presented with parkinsonism (bradykinesia and rigidity), and five presented with supranuclear gaze palsy or ophthalmoplegia. Because this disease tends to cause cerebellar ataxia, it is also a differential diagnosis in patients with PSP and cerebellar ataxia (PSP-C).

One exemplar case report presented the case of a 45-year-old woman with breast cancer [37] who presented with subacute progression of PSP-like manifestations, including gait bradykinesia, gait freezing, dysphagia, urinary urgency, complete external ophthalmoplegia (both horizontally and vertically), and axial rigidity. However, symptoms were atypical for PSP, as the patient also presented with leg spasticity with no MRI abnormalities characteristic of PSP. The patient was treated with IVIG and chemotherapy; however, the treatment efficacy was only temporary.

### 3.5. Anti-CV2/CRMP5 Antibody

#### 3.5.1. General Features of Anti-CV2/CRMP5 Antibody

The anti-CV2/CRMP5 antibody recognizes CRMP5, an intracellular antigen [47] primarily expressed in the brain and spinal cord whose functions include neuronal migration, axonal guidance, and dendritic growth.

Regarding diseases associated with anti-CV2/CRMP5 antibodies, there is a high risk of tumor coexistence (>80%), frequently including thymoma, a generally benign anterior mediastinal tumor, and SCLC, an aggressive form of lung cancer [44]. This condition typically occurs in a subacute fashion, manifesting as encephalomyelitis (a term that should only be used to describe patients presenting with multiple neurological manifestations including limbic encephalitis, brainstem encephalitis, cerebellar degeneration, myelitis, sensory neuronopathy, and chronic gastrointestinal pseudo-obstruction [44,48]), neuropathy (primarily asymmetric polyradiculoneuropathy), chorea, or ocular manifestations (mainly optic neuritis, retinitis, and uveitis) [47,49,50,51]. When suspected, immunoblotting can be performed for antibody detection. The effects of immunotherapy and tumor treatment are limited [47,51].

#### 3.5.2. The Role of Anti-CV2/CRMP5 Antibody in PSP-like Manifestations

Only one case report has yet discussed the relationship between anti-CV2/CRMP5 antibody and PSP [32]. A 65-year-old man with SCLC had subacute onset of the disease, with PSP-like symptoms and signs, including vertical supranuclear gaze palsy, bradykinesia, rigidity, postural instability, and personality change. Symptoms not typically present in PSP included a weight loss of 10 kg over 6 months and T2 hyperintensity in the basal ganglia on magnetic resonance imaging (MRI). Intravenous immunoglobulin (IVIG) and chemotherapy were effective.

### 3.6. Anti-Hu Antibody

#### 3.6.1. General Features of Anti-Hu Antibody

Anti-Hu antibodies act against intracellular antigens belonging to the Hu family, particularly embryonic-lethal-abnormal-vision-like 4 (ELAVL4), which is primarily expressed in the brain and plays a role in the binding and stabilization of mRNA. Diseases associated with anti-Hu antibodies are at high risk (85%) of tumor co-existence, most commonly SCLC [44]. The typical clinical presentations include subacute sensory neuropathy, limbic encephalitis, and cerebellar ataxia. Immunotherapy and oncological therapy are only partially effective [52].

#### 3.6.2. The Role of Anti-Hu Antibody in PSP-like Manifestations

To date, only one patient with PSP-like symptoms has been reported [33]. This patient was a 57-year-old man with SCLC who had a subacute disease course. His symptoms, also common in PSP, included supranuclear vertical gaze palsy, whereas the symptoms atypical of PSP included T2 hyperintensity in the extreme capsule on MRI and elevated protein and IgG index in the CSF. The patient was treated with IVIG in a single cycle (400 mg/kg/dose for 5 days), chemotherapy, and radiotherapy, all of which were ineffective.

## 4. Diagnosis of AE/PNS Mimicking PSP

### 4.1. Suspecting AE/PNS Mimicking PSP

From the perspective that it is a treatable condition, it is essential to actively consider the possibility of AE/PNS when encountering PSP-like clinical symptoms. AE/PNS should be suspected in patients with manifestations atypical for PSP (Table 3). The key diagnostic clues are as follows: young age of onset (<40 years of age), acute or subacute disease course, coexisting neoplasms, CSF abnormalities (elevated protein, pleocytosis, increased IgG index, and positive CSF-specific OCB), and brain MRI with no findings typical of PSP (e.g., atrophy of the midbrain tegmentum or superior cerebellar peduncle). It is also important to note the characteristic findings of each antibody, which include significant sleep disorders, behavioral manifestations, respiratory failure, and orthostatic hypotension in anti-IgLON5 disease; narcolepsy in the presence of anti-Ma2 antibody; and diarrhea and severe weight loss in diseases associated with anti-DPPX antibody [11].

### 4.2. Confirmation of Diagnosis

#### 4.2.1. General Considerations

If an AE/PNS mimicking PSP is suspected, laboratory assessments and considerations of clinical information are necessary for establishing the diagnosis.

#### 4.2.2. Laboratory Assessments

Laboratory assessments can be achieved in two steps: a tissue-based assay (TBA) and a confirmatory test. One of two different confirmatory tests can be performed, depending on the target antibodies. One is CBA, and the other is immunoblotting. Although the results of a TBA and a confirmatory test should essentially be concordant, if this is not the case, the sensitivity and specificity of each test must be considered [44,53,54,55,56].

#### 4.2.3. Considerations of Clinical Information

In addition to laboratory results, it is often necessary to confirm that the clinical course is typical [44,57]. However, the number of reported patients with AE/PNS mimicking PSP is not sufficiently large to establish these PSP-like manifestations as true clinical entities, particularly for antibodies other than anti-IgLON5. Therefore, careful differential diagnosis is important. When the results are uncertain, first-line treatment can be considered for diagnostic purposes.

## 5. Management of AE/PNS Mimicking PSP

### 5.1. General Principles

Immunotherapy and the management of neoplasms are the mainstays of treatment. As there is no treatment with a high level of evidence for AE/PNS mimicking PSP, commonly recommended treatments for AE/PNS are typically followed [58,59,60,61]. Immunotherapy can be divided into acute and maintenance therapies.

### 5.2. Acute Immunotherapies

#### 5.2.1. First-Line Therapies

The first-line acute therapy is often intravenous methylprednisolone (IVMP), either alone or in combination with IVIG or plasmapheresis. The regimen for a single cycle comprises 1000 mg of intravenous methylprednisolone for three days for IVMP; 2 g/kg divided into five days for IVIG; and five to ten sessions of therapeutic plasma exchange every alternate day for plasmapheresis.

#### 5.2.2. Second-Line Therapies

If these treatments are deemed inadequate at follow-up performed two to four weeks following the completion of acute treatment, RTX or cyclophosphamide is considered as a second-line therapy [59]. However, it is crucial to re-evaluate the diagnosis before initiating these therapies, particularly when the first-line therapies are completely ineffective or laboratory tests are indeterminate. RTX can be administered at a dose of 375 mg/m^2^ four times weekly, whereas cyclophosphamide can be administered at 600–1000 mg/m^2^ monthly for up to six months.

### 5.3. Maintenance Immunotherapies

In cases without recurrence, oral corticosteroids can be prescribed as a bridging therapy, with tapering over several months; long-term maintenance therapy is not common. On the other hand, maintenance therapy can be considered in patients who test positive for cell-surface antibodies and experience recurrence [60,61]. An important point to note is that a higher level of diagnostic uncertainty in patients with AE/PNS mimicking PSP necessitates a thorough differential diagnosis before starting maintenance therapies.

When conducting maintenance therapies, the common regimen includes either periodic RTX alone, or azathioprine (AZA) or mycophenolate mofetil (MMF) with oral corticosteroids tapered over three to six months as a bridging therapy. Periodic RTX can also be administered at a dose of 375 mg/m^2^ every four to six months. AZA can be administered initially at a dose of 1.5 mg/kg/day titrated to 2–3 mg/kg/day over one to several months, while MMF is initially dosed at 500 mg twice daily, increasing to 1000 mg twice daily after two weeks. The duration of maintenance treatment can initially be considered as three years [60].

### 5.4. Tumor Therapies

Coexisting neoplasms should be treated simultaneously. If a related tumor is detected, complete resection can be initially considered. If this is not possible, debulking surgery, chemotherapy, or radiotherapy should be considered.

## 6. AE/PNS Associated with Other Atypical Parkinsonism

AE/PNS can also present with manifestations resembling those of corticobasal syndrome (CBS) and multiple system atrophy (MSA).

First, in CBS-like cases, when limiting to those with multiple reported patients, anti-IgLON5 (two patients) [16,62] and anti-glutamic acid decarboxylase (GAD) antibodies (four patients) [63,64] have been documented. The range of age was 49 to 78 years, and all cases exhibited chronic progression. Key factors for differential diagnosis from neurodegenerative CBS included CSF abnormalities throughout all the antibody types; dystonia-like muscle stiffness, muscle spasms, and paraspinal muscle hypertrophy in the diseases associated with anti-GAD antibody; and leg-predominant apraxia in the ones related to anti-IgLON5 antibody. The majority of these patients had effective immunotherapy.

Next, as for MSA-mimicking diseases, also focusing on those with two or more reported patients, anti-contactin-associated protein 2 (Caspr2) (two patients) [65], anti-CV2/CRMP5 (two patients) [66,67], and anti-Homer-3 antibodies (two patients) [68] were identified. The age ranged from 48 to 72 years. Differentiating factors from MSA across all these antibodies included subacute-to-acute clinical progression and CSF abnormalities. Specific distinguishing features associated with anti-Caspr2 antibodies included a variety of symptoms such as hyponatremia, neuropathy, encephalopathy, hallucinations, and epilepsy, while no distinctive features were reported for anti-CV2/CRMP5 and anti-Homer-3 antibodies. Immunotherapy was effective in most cases.

## 7. Conclusions and Future Directions

With advances in antibody detection technologies, autoimmune parkinsonism has become attractive as a treatable condition. When we encounter a patient with PSP-like syndromes, the possibility of an autoimmune disease should always be considered, rather than simply assuming that the symptoms are caused by PSP as a neurodegenerative disease. However, as yet, the association between AE/PNS and PSP-like manifestations has not been adequately proven. Antibody detection assays unfortunately have a relatively high risk of presenting with false-positive and false-negative results. Moreover, only a single or handful of PSP-like patients have been described in detail for all antibodies, except for the anti-IgLON5 antibody. The accumulation of patients with similar clinical courses is essential to overcome these problems.

## Figures and Tables

**Table 1 brainsci-14-01012-t001:** Brief overview of MDS-PSP criteria.

**Mandatory inclusion criteria** (1) Sporadic occurrence (2) Age 40 or older at onset of first PSP-related symptom (3) Gradual progression of PSP-related symptoms **Mandatory exclusion criteria** (e.g., episodic memory dysfunction, prominent autonomic failure, visual hallucinations, upper and lower motor neuron signs, sudden onset or stepwise or rapid progression, prominent appendicular ataxia, severe leukoencephalopathy and relevant structural abnormality (e.g., normal pressure hydrocephalus) on imaging)
**Context-dependent exclusion criteria** (e.g., CADASIL or prion disease for patients with relevant imaging findings, AD pathology for PSP-CBS patients, Wilson’s disease, Niemann–Pick disease, type C, hypoparathyroidism, neuroacanthocytosis, and neurosyphilis for ones aged <45, prion disease or paraneoplastic encephalitis for ones with rapid progression, genetic causes (e.g., *MAPT*) for ones with positive family history)
**Core clinical features** Ocular motor dysfunction: O1, vertical supranuclear gaze palsy; O2, slow velocity of vertical saccades; O3, frequent macro square wave jerks of eyelid opening apraxia. Postural instability: P1, repeated unprovoked falls within 3 years; P2, tendency to fall on the pull-test within 3 years; P3, more than two steps backward on the pull-test within 3 years. Akinesia: A1, progressive gait freezing within 3 years; A2, parkinsonism, akinetic-rigid, predominantly axial, and levodopa-resistant; A3, parkinsonism, with tremor and/or asymmetric and/or levodopa response. Cognitive dysfunction: C1, speech/language disorder, i.e., nonfluent/agrammatic variant of primary progressive aphasia or progressive apraxia of speech; C2, frontal cognitive/behavioral presentation; C3, corticobasal syndrome. Clinical clues: CC1, levodopa resistance; CC2, hypokinetic, spastic dysarthria; CC3, dysphagia; CC4, photophobia.
**Diagnostic criteria *** Definite PSP: gold standard defining the disease entity which is accomplished by neuropathological diagnosis. Probable PSP: probable PSP-RS, (O1 or O2) + (P1 or P2); probable PSP-PGF, (O1 or O2) + A1; probable PSP-P, (O1 or O2) + (A2 or A3); probable PSP-F, (O1 or O2) + C2. Possible PSP: possible PSP-OM, O1; possible PSP-RS, O2 + P3; possible PSP-PGF, A1; possible PSP-SL, (O1 or O2) + C1; possible PSP-CBS, (O1 or O2) + C3. Suggestive of PSP: s.o. PSP-OM, O2 or O3; s.o. PSP-PI, P1 or P2; s.o. PSP-RS, O3 + (P2 or P3); s.o. PSP-P, (A2 or A3) + (O3, P1, P2, C1, C2, CC1, CC2, CC3, or CC4); s.o. PSP-SL, C1; s.o. PSP-F, C2 + (O3 or P3); s.o. PSP-CBS, C3.

* Mandatory inclusion and exclusion criteria and context-dependent exclusion criteria apply for all probable, possible, and suggestive PSP criteria. Abbreviations: Alzheimer’s disease, AD; cerebral autosomal dominant arteriopathy with subcortical infarcts and leukoencephalopathy, CADASIL; Movement Disorder Society clinical diagnostic criteria for progressive supranuclear palsy, MDS-PSP criteria; progressive supranuclear palsy, PSP; PSP-corticobasal syndrome, PSP-CBS; PSP-frontal presentation, PSP-F; PSP-ocular manifestation, PSP-OM; PSP-parkinsonism, PSP-P; PSP-progressive gait freezing, PSP-PGF; PSP-Richardson syndrome, PSP-RS; PSP-speech/language disorder, PSP-SL.

**Table 3 brainsci-14-01012-t003:** Clinical features suggestive of AE/PNS mimicking PSP.

(1) Younger age of onset (<40 years of age) (2) Acute or subacute progression (3) Concurrent neoplasms (4) Abnormal CSF (i.e., elevated protein, pleocytosis, increased IgG index, and positive CSF-specific OCB) (5) No MRI findings suggestive of PSP (6) Antibody-specific manifestations atypical for PSP (i.e., significant sleep disorder, behavioral alterations, respiratory failure, and orthostatic hypotension for anti-IgLON5 antibody; and narcolepsy for anti-Ma2 antibody) (7) Other symptoms and signs much less common in PSP (e.g., prominent weight loss, sleep disturbance, and leg ataxia)

Abbreviations: autoimmune encephalitis or paraneoplastic neurological syndromes, AE/PNS; cerebrospinal fluid, CSF; magnetic resonance imaging, MRI; oligoclonal band, OCB; progressive supranuclear palsy, PSP.

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
