# Peer review of "Autoimmune Encephalitis and Paraneoplastic Neurological Syndromes with Progressive Supranuclear Palsy-like Manifestations"

_brainsci, 2024, doi:10.3390/brainsci14101012_

Round 1
Reviewer 1 Report
Comments and Suggestions for Authors
1. The abstract does not clearly define the objective of the study. It is in Line 46 under introduction that the authors describe the purpose of the study for the first time. Here also the authors could be more precise and descriptive as the objective of the study is to assess the antibodies associated with AE/PNS related PSPs.
2. Lines 53-58 has asterisk annotations either replacing an alphabet or as an additional sign, eg 'y' in Palsy, line 53 is replaced with * and similarly 'y' in the word antibody in line 56 is replaced with the asterisk.
3. Line 79 use of 'a neurodegenerative disease' is redundant here and will be well suited if written in the introduction.
4. Although the summary on anti-Iglon5 antibody in line 93 states that CSF analysis is sensitive for these, the 3 references describing same antibody in table 1, Gonzalez-Avila et al., 2020; Gaig et al., 2021 and Berger Sieczkowski et al., 2023 either indicate that abnormal CSF findings were N/A or were normal. Thus it is unclear whether these antibodies are detected in CSF or not.
5. The authors should briefly describe the difference in PSP, PSP-Richardson symptom and PSP-corticobasal syndrome under 3.2.2
6. Based on the linse 103-119 although most studies on PSP which tested for anti-IgLON5 showed negative correlation between PSP and the antibodies, patients with anti-IgLON5 disease have repeatedly shown PSP like manifestations. Therefore are the authors suggesting that although these antibodies cannot be implicated as cause of PSP there is reasonable evidence for anti-IgLON5 neurological disorder manifesting as a PSP? Thus 3.2.2 will benefit from a conclusive remark at the end.
7. Line 121 under 3.2.2 needs brief description on' 3-repeat' and '4-repeat tauopathy'. Line 123 under 3.2.2 needs some context on role of tufted astrocytes in PSP. Line 126, needs explnation of 'neuropil threads'
8. Grammatical error in line 158 under 3.4.1
9. Authors should add brief description of thymoma and SCLC in line 162
10. Authors should describe IVIG in line 193
11. Title of Table 2 needs grammatical correction.
12. Point 7 in table 2 needs explanation with examples.
12. Section 4.2.2, the title and first word of the paragraph should be full form of the word instead of abbreviation.
13. Lines 371-381 needs appropriate references and rephrasing. These lines currently sound as if the authors are prescribing treatment regimen for the respective cases and not describing the current regimes followed in the clinical settings.
14. The structure of section 3 can be improved and made more coherent by combining the sub sections describing summary of a given antibody and its role in PSP.
Comments on the Quality of English LanguageThe quality of English is above average and the paper has few grammatical errors which could be easily corrected.
Reviewer 2 Report
Comments and Suggestions for Authors
Yamahara et al. present a review of autoimmune encephalitis (AE) and paraneoplastic neurological syndromes (PNS) manifesting with symptoms of progressive supranuclear palsy (PSP). Considering the rarity of these conditions, a focus on this phenotype is indeed of interest.
I have some minor comments/suggestions:
- In the introduction, a box reporting the main diagnostic criteria for PSP according to the Movement Disorder Society could be of help to better highlight the features that are proper to the condition. Also, additional details on the pathogenesis, instrumental features and treatment of PSP should be added to enable to reader to better appreciate the differences existing between this degenerative disorder and immune-mediated PSP-like conditions.
- In the methods section, the authors should define which are the "PSP-like manifestation" considered to paper selection; a list would be recommended and if one feature was sufficient or whether more were required;
- In the main body of the text (paragraph 3), I would suggest to widen the description of PSP-like phenotypes associated with IgLON5 antibodies, since this condition is the one more recurrently associated with the PSP phenotype; clinical clues that could be detected to distinguish degenerative PSP from immune-mediated PSP-like phenotypes should be especially emphasized. Antibody types for which a single case has been associated with a PSP-like phenotype could perhaps only appear in the table.
- In Table 1, I suggest to add an additional column that better decsribes the PSP-like features identified in patients; I would also suggest to group cases in order of antibody type to ease the reader;
- I would suggest to emphasize in paragraph 4 the importance of recognizing these immune-mediated conditions considering that, differently from PSP, treatments are available and can produce some improvement
- In paragraph 4, I would strongly suggest to remove the subsections on TBA, CBA and immunoblot that contain a number of imprecise information.
Reviewer 3 Report
Comments and Suggestions for Authors
In a review by Yahamara et al, authors analyze the associations between the autoimmune encephalitis, paraneoplastic neurological syndromes and Progressive Supranuclear Palsy-like manifestations. I have the following comments regarding this work:
1. Authors should emphasize the multiple theories regarding the pathogenesis of PSP, which is certain aspects overlaps with neoplasms for example in the context of chronic inflammation. Recently the significance of this factor in PSP was examined using the associations between neuroimaging and interleukins among others.
2. The PSP syndrome should be highlighted as a group of clinical entities rather than a single disease which may have mimicking manifestations – Ref. Differential Diagnosis of Rare Subtypes of Progressive Supranuclear Palsy and PSP-Like Syndromes-Infrequent Manifestations of the Most Common Form of Atypical Parkinsonism. Front Aging Neurosci. 2022 Feb 9;14:804385. doi: 10.3389/fnagi.2022.804385. PMID: 35221993; PMCID: PMC8864174. / PSP-Richardson syndrome mimics: An overview and pragmatic approach. Rev Neurol (Paris). 2024 Jan-Feb;180(1-2):12-23. doi: 10.1016/j.neurol.2023.02.070. Epub 2023 Aug 3. PMID: 37543508.
3. The review concerning the types of paraneoplastic syndromes lacks a critical perspective highlighting the strengths and weaknesses of the studies. In its current version it is more a list of studies with a short overview. Authors should consider additionally grouping works based on the examinations of single patients as the significance of their outcome may seem questionable.
4. The section concerning treatment should be more structured.
5. The perspective on PSP-mimicking diseases could be deepened by elaboration on its stages.
6. I assume a paragraph which is clearly linked with differential diagnosis of PSP and PSP-mimicking diseases would be valuable.
7. It would be interesting to provide a section regarding possible associations of autoimmune encephalitis and paraneoplastic neurological syndromes with other less common atypical parkinsonisms e.g. Corticobasal Syndrome
Round 2
Reviewer 3 Report
Comments and Suggestions for Authors
I do not have further comments.